# Endogenous clock-mediated regulation of intracellular oxygen dynamics is essential for diazotrophic growth of unicellular cyanobacteria

Anindita Bandyopadhyay [1], Annesha Sengupta [1,3], Thanura Elvitigala [1,2] & Himadri B. Pakrasi [1] ✉

The discovery of nitrogen fixation in unicellular cyanobacteria provided the first clues for the existence of a circadian clock in prokaryotes. However, recalcitrance to genetic manipulation barred their use as model systems for deciphering the clock function. Here, we explore the circadian clock in the now genetically amenable *Cyanothece* 51142, a unicellular, nitrogen-fixing cyanobacterium. Unlike non-diazotrophic clock models, *Cyanothece* 51142 exhibits conspicuous self-sustained rhythms in various discernable phenotypes, offering a platform to directly study the effects of the clock on the physiology of an organism. Deletion of *kaiA*, an essential clock component in the cyanobacterial system, impacted the regulation of oxygen cycling and hindered nitrogenase activity. Our findings imply a role for the KaiA component of the clock in regulating the intracellular oxygen dynamics in unicellular diazotrophic cyanobacteria and suggest that its addition to the KaiBC clock was likely an adaptive strategy that ensured optimal nitrogen fixation as microbes evolved from an anaerobic to an aerobic atmosphere under nitrogen constraints.

Living organisms constantly need to adapt to rhythmic changes in their environment that are generated by the Earth's rotation. To adapt to such constantly changing conditions, most organisms have evolved to develop an endogenous time-keeping system that proactively synchronizes physiological processes to the external light-dark cycle. Over the years, studies have established intimate connections between this time-keeping clock and cellular metabolism across all phylogenetic groups, with the endogenous rhythms orchestrating key parameters like growth, development, stress response, and reproductive fitness[1–3].

Concerted efforts over decades have unraveled various facets of the circadian clock in different model systems. The simplest model organisms to possess a bona fide and fully functional internal clock are cyanobacteria, oxygenic photosynthetic prokaryotes[1,4,5]. Cyanobacteria have inhabited our planet for billions of years and have successfully transitioned through the extremely harsh environmental and geophysical conditions that prevailed during different phases of Earth's evolutionary history. Some of the key events that shaped Earth's early biome and its evolution thereafter include the nitrogen crisis, which led to the emergence of nitrogen-fixing organisms[6,7] and cyanobacteria-mediated transformation of Earth's initial reducing geochemical environment into an oxidizing one[8,9] that resulted in the burgeoning of aerobic life on our planet. Moreover, due to various geophysical events, the earth's rotation period changed significantly over the history of our planet, subjecting evolving microbes to major day-length changes[10]. Central to the coping mechanisms deemed

[1]Department of Biology, Washington University, St. Louis, MO, USA. [2]General Motors Research and Development, Warren, MI 48092, USA. [3]Present address: Department of Chemical Engineering, University of Toronto, Toronto, ON, Canada. ✉e-mail: pakrasi@wustl.edu

necessary for survival and fitness under these changing circumstances, was the evolution of a timekeeping mechanism that would orchestrate the physiological processes of microbes in response to the environment[4,9].

The observation made decades ago, that unicellular cyanobacteria can perform nitrogen fixation under continuous light, hinted at the existence of an endogenous rhythm that segregates photosynthesis from nitrogen fixation[11,12]. These initial studies provided the first clues for the occurrence of a circadian clock in prokaryotes. Subsequent studies in unicellular diazotrophic cyanobacteria suggested a role for the endogenous clock in orchestrating not only nitrogen fixation but other metabolic processes as well[13,14]. However, owing to their recalcitrance to any genetic manipulation, the clock-mediated segregation of processes remained largely unexplored in this group of prokaryotes. Instead, the unicellular non-diazotrophic cyanobacteria *Synechococcus elongatus* PCC 7942 was adopted as the model system for understanding the cyanobacterial clock mechanism. Many elegant studies in this strain over the past decades have unraveled various facets of the cyanobacterial clock. A genetic screen identified a cluster of three genes *kaiA*, *kaiB*, and *kaiC*, that encode the core proteins of the clock[15]. KaiA promotes the autokinase activity of KaiC while KaiB promotes its intrinsic autophosphatase activity by inhibiting the stimulatory effect of KaiA[16,17]. Subsequent studies demonstrated the role of the clock in regulating global patterns of gene expression and cell division[18,19]. The fundamental oscillatory mechanism of the clock was also deciphered[20,21]. However, a direct involvement of the clock in regulating physiological processes could not be demonstrated[5].

Studies have extensively investigated the temporal separation of photosynthesis and nitrogen fixation in unicellular diazotrophic strains like *Crocosphaera* and *Cyanothece* under diel growth conditions[22–27]. Studies have also shown nitrogen fixation under continuous light and in the subjective dark period of light/dark entrained cells[27–29]. However, the role of the clock in orchestrating these processes, though hypothesized by many, has not been studied under any of these conditions. Here we provide the first direct evidence of the involvement of the circadian clock in maintaining endogenous rhythms of physiological processes in a unicellular diazotrophic cyanobacteria that performs aerobic nitrogen fixation. Our work shows that *kaiA*, a clock gene that occurs only in cyanobacteria among all prokaryotes, is required for maintaining robust rhythms in intracellular oxygen cycling, a phenomenon that provides a stringently controlled cellular environment for nitrogen fixation in an otherwise oxygen-rich platform. Abolishing the rhythms, as is observed in the Δ*kaiA* mutant, negatively impacts nitrogen fixation rates which can be revived by providing an anaerobic environment to the cells. Our findings suggest that a *kaiA*-controlled circadian clock is essential for optimal growth under diazotrophic conditions and such a clock likely

evolved as nitrogen-fixing cyanobacteria transitioned from an anaerobic to an aerobic atmosphere.

## Results

### The clock is indispensable in *Cyanothece* 51142

Analysis of the genome of *Cyanothece* 51142 revealed that the canonical *kaiA* (cce_0424), *kaiB1* (cce_0423), and *kaiC1* (cce_0422) genes that are ubiquitous among cyanobacteria, are part of a larger cluster that is well conserved within unicellular diazotrophic cyanobacteria that fix nitrogen aerobically[26,30,31] (Supplementary Fig. 1). In order to decipher the role of the KaiABC clock, we used CRISPR to create deletion mutants of *kaiA*, *B1* and *C1* genes in *Cyanothece* 51142. Under nitrogen sufficient conditions, we could obtain completely segregated mutants of *kaiA* and *kaiB1* but the *kaiC1* CRISPR mutant did not segregate in our hands (Supplementary Fig. 2). Therefore, we attempted to delete the *kaiC1* gene by the more conventional homologous gene replacement method and enforced segregation of the transgenic lines by applying antibiotic pressure. Even under very high antibiotic pressure a WT copy of the *kaiC1* gene was retained in the genome. We then attempted to eliminate the *kaiC1* gene by deleting the entire *kaiAB1C1* cluster, an approach that also remained unsuccessful. Our repeated failure to delete the *kaiC1* gene indicated that unlike *Synechococcus* 7942, where all the clock genes are dispensable[32,33], the clock cannot be completely eliminated in *Cyanothece* 51142. Comparison of the growth phenotypes of the Δ*kaiA* and Δ*kaiB1* mutants under CL in the absence of any added nitrogen sources revealed that of the two, *kaiA* is essential for sustenance of robust endogenous rhythms observed in the WT (Supplementary Fig. 3). To investigate the role of these endogenous rhythms in unicellular diazotrophic cyanobacteria, in this study we focus on the Δ*kaiA* mutant of *Cyanothece* 51142.

### The Δ*kaiA* mutant exhibits various phenotypic defects

To evaluate the effect of the deletion of *kaiA* on the physiology of *Cyanothece* 51142, we compared the growth of the WT and the Δ*kaiA* mutant under continuous light (CL) and light-dark (LD) conditions (photoautotrophic growth), in media lacking any fixed sources of nitrogen. When grown under CL and nitrogen-fixing conditions, in the absence of any external entraining light cues, the WT exhibits an endogenous rhythm of approximately 24 h (~16:8 LD phases) (Supplementary Fig. 3, Fig. 1a). The rhythm corresponds to a rise in OD 730 during the photosynthetic phase of growth (approximately 16 h), followed by a drop in OD 730 for the next 8 hours of the nitrogen-fixing phase. In contrast to the WT, the Δ*kaiA* mutant exhibited a significantly dampened rhythm under CL, which eventually faded away with time. The rhythms in the growth of the WT and the Δ*kaiA* mutant were largely consistent across multiple runs (Source data). Multiple colonies of the mutant were tested (see method section) and consistency in rhythm was also observed across all the tested clones. To assess if the

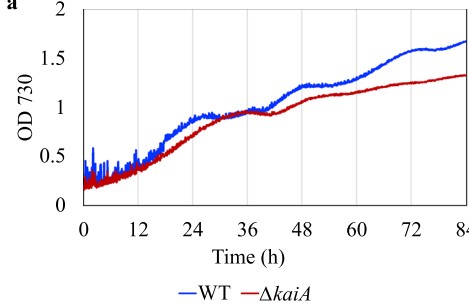

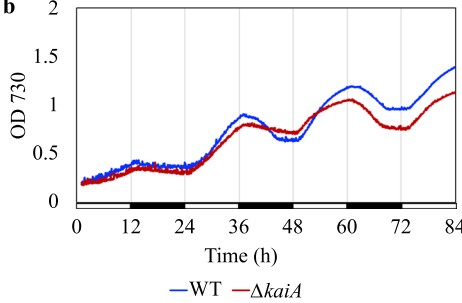

**Fig. 1 | Growth comparison of the WT and Δ*kaiA* strains of *Cyanothece* 51142 in nitrogen-deficient media.** Self-sustained endogenous rhythms of ~24 h are reflected in growth under CL(**a**). Rhythm imposed by 12:12 LD cycles (**b**). The CL curve is an average of three independent runs of the WT and 5 independent runs of the mutant. The LD curve is an average of 3 independent runs of the WT and the mutant. Black bars represent dark phase of growth.

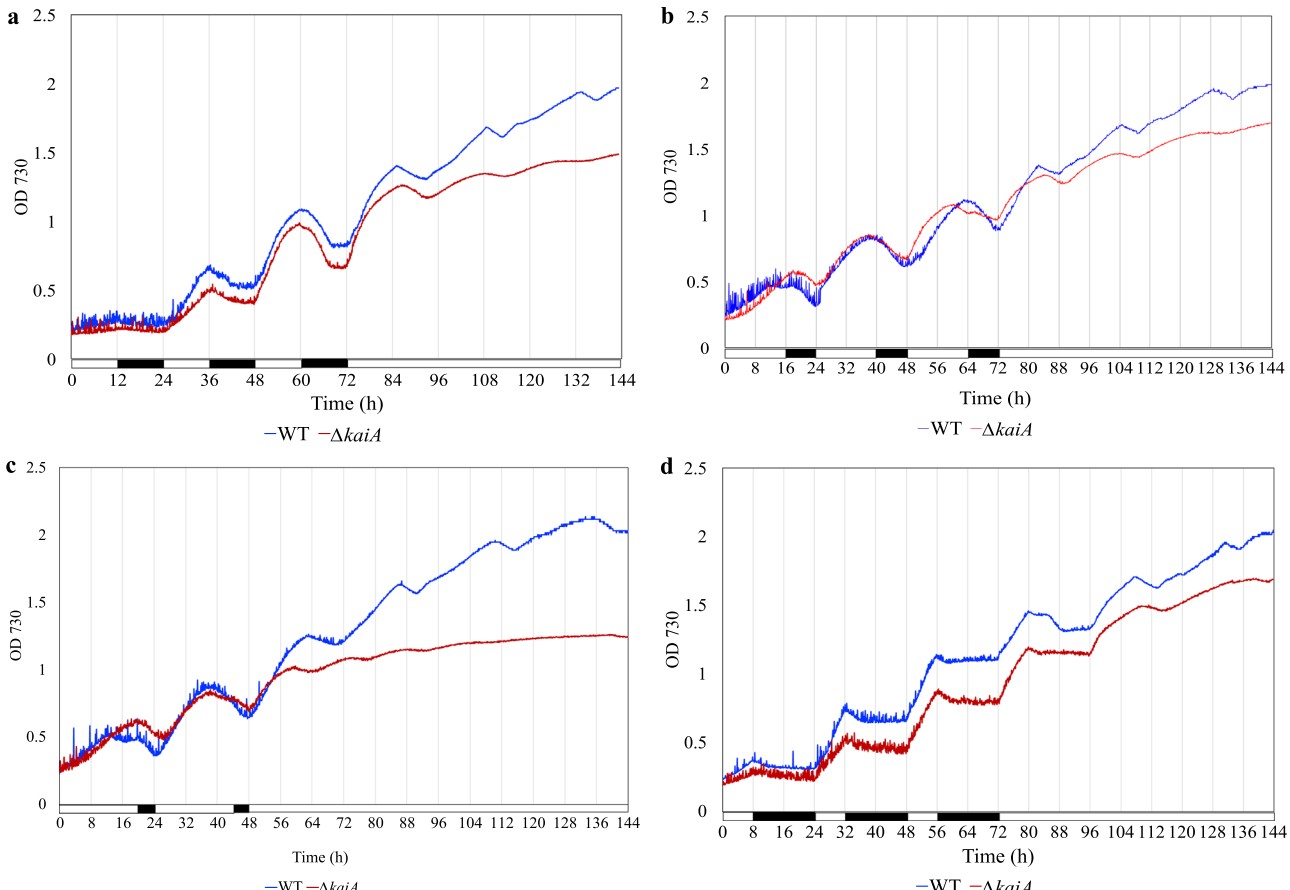

**Fig. 2 | Sustained endogenous rhythm of growth in *Cyanothece* 51142.** Representative curves showing a comparison of the endogenous rhythms of the WT and Δ*kaiA* strains grown under different LD regimes – 12:12 (**a**), 16:8 (**b**), 20:4 (**c**), and 8:16 (**d**) followed by CL growth. Representative growth curves from multiple runs of the WT and mutant are presented. The black bar represents a dark phase of growth.

loss in rhythm observed in the mutant was a direct effect of the deletion of *kaiA* and not due to a malfunctioning of the remaining clock components, we generated a Δ*kaiA'* complementation strain (Supplementary Fig. 4a). The complementation helped restore the rhythms under CL to WT levels, confirming that the loss of rhythm in the mutant can solely be attributed to the absence of KaiA (Supplementary Fig. 4b).

To enhance the visibility of the cyclic patterns of the growth data as well as to perform quantitative analysis, small fluctuations in the growth measurements were removed using a moving-average filter. Then the overall growth trend of the samples was approximated by fitting an exponential curve. We then subtracted these trends from the smoothed growth measurements to elucidate the cyclic patterns in the data. Supplementary Fig. 5 shows the detrended growth of three independent runs (measured using OD 730) of the WT and five independent runs of the Δ*kaiA* mutant over 4 days. Our analysis showed that the WT strain not only exhibited greater oscillations compared to the mutant, it also maintained these patterns over the full duration of the experiment. Cyclic variations in the growth of the mutant dampened quickly. In order to quantitatively compare the diurnal patterns observed in the CL growth data, we performed the power spectrum density (PSD) analysis on the (detrended) growth data. Supplementary Fig. 6, shows the PSD of growth data at different frequencies of a selected sample of the WT and a Δ*kaiA* mutant. The analysis revealed that the WT strain had an overall higher level of PSD compared to the mutant. It also revealed that the PSD of the wild type at the diurnal frequency (that is 1 cycle/day) was more than twice that of the mutant.

We employed the Mann-Whitney U-test to compare the growth parameters and the diurnal oscillations between WT and the Δ*kaiA* mutants. This analysis revealed a statistically significant ($p$-value < 5%) higher level of growth as well as larger diurnal oscillations in the WT compared to the Δ*kaiA* mutant.

The Δ*kaiA* mutant also exhibited significantly reduced biomass accumulation, reduced chlorophyll content, and a bleached phenotype which correlated with the observed degradation in phycocyanin levels (Supplementary Fig. 7, S8a, c). Under 12 h LD and diazotrophic growth conditions, the endogenous rhythm in growth coincides with the imposed LD cycles, with OD increasing for the first 12 h of light followed by a decrease in OD for the subsequent 12 h of darkness. This decrease in OD during the dark phase reflects carbohydrate degradation and utilization during nitrogen fixation[34]. The mutant exhibited a similar diurnal rhythm in growth pattern as the WT although biomass accumulation was lower (Fig. 1b, Supplementary Fig. 7). Absorption spectra of samples grown under 12:12LD conditions revealed a slight reduction in phycocyanin content (also reflected in the color of the culture) but chlorophyll content was not significantly different (Supplementary Fig. 8b, d).

An intrinsic property of the circadian clock is its ability to sustain the rhythmicity imposed by an external cue, even in the absence of the cue. To assess the role of KaiA in sustaining the endogenous periodicity, we compared the growth of the WT and the Δ*kaiA* mutant subjected to different light-dark regimes followed by growth under CL (Fig. 2). When cells grown under different LD cycles (12:12, 16:8, 20:4 and 8:16) were transitioned to CL, the WT grown under any of the above conditions, maintained a strong periodicity for several days. In

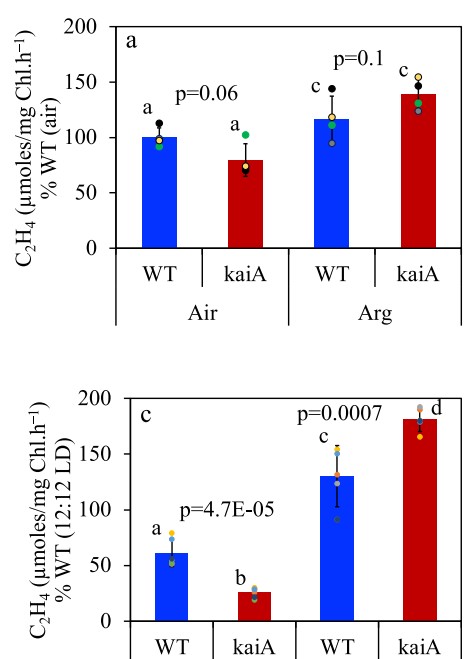

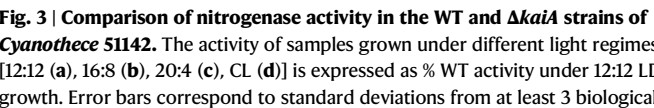

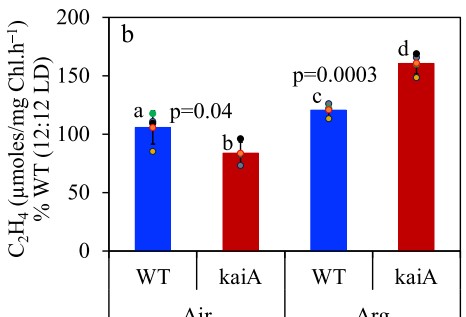

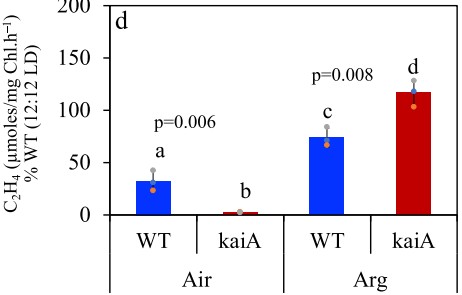

**Fig. 3 | Comparison of nitrogenase activity in the WT and Δ*kaiA* strains of *Cyanothece* 51142.** The activity of samples grown under different light regimes [12:12 (**a**), 16:8 (**b**), 20:4 (**c**), CL (**d**)] is expressed as % WT activity under 12:12 LD growth. Error bars correspond to standard deviations from at least 3 biological replicates (12:12 LD, *n* = 4), (16:8 LD, *n* = 5) and (20:4 LD and CL, *n* = 3) The letters **a**, **b** denote statistically different values of μ for Air and **c**, **d** indicate statistical significance for Argon (*p* < 0.05), while same letters a or c indicate that the difference in values of μ is insignificant (*p* ≥ 0.05).

contrast, the oscillations observed in the mutant dampened significantly over time under CL and biomass accumulation was also significantly lower (Fig. 2a–d, Supplementary Fig. 9). Although reduction in biomass in the mutant was a common observation upon subjecting cells entrained under various LD conditions to CL, across multiple growth experiments various arrhythmic growth patterns were also observed, particularly under more extreme light regimes (20:4 LD and 8:16 LD) (Supplementary Fig. 10).

**Deletion of *kaiA* affects nitrogenase activity**
To assess if the KaiA-controlled clock plays a role in the nitrogen fixation process in *Cyanothece* 51142, we compared nitrogenase activity in the WT and Δ*kaiA* strains grown under different day-length conditions. When cells grown under 12:12 h or 16:8 h LD cycles were incubated under aerobic conditions, the average specific rates of nitrogenase activity in the Δ*kaiA* mutant were not significantly different from that of the WT (Fig. 3a,b, Supplementary Table 1). Under 20:4 h LD and CL, the WT exhibited reduced rates of nitrogenase activity as compared to under 12:12 h LD growth conditions (Fig. 3c,d). Nitrogen fixation rates in the mutant were greatly reduced (∽30% and 5% of WT) when cells grown under these conditions were aerobically incubated. Interestingly, when cells grown under longer day length conditions (16:8, 20:4, and CL) were incubated anaerobically, the Δ*kaiA* mutant exhibited significantly higher rates of nitrogenase activity compared to the WT (Fig. 3). To assess if the reduction in nitrogenase activity in Δ*kaiA* under aerobic incubation conditions was a result of reduced transcript levels of the nitrogenase genes, we performed qRT-PCR analysis of a few representative genes of the *nif* cluster. To our surprise, we observed significant upregulation of the *nif* genes (4 to 10 fold) in the Δ*kaiA* samples collected from CL or LD (L2, D2, see methods) growth conditions (Supplementary Fig. 11a). Analysis of transcript levels of genes involved in respiration, a process critical for nitrogen fixation also revealed significant upregulation in the Δ*kaiA* mutant (Supplementary Fig. 11b). Under 8:16 LD cycles, nitrogenase activity was significantly low in the WT compared to 12:12 LD

conditions and the Δ*kaiA* mutant showed rates ∽20% of the WT (Supplementary Fig. 12). Mutant cells grown under 8:16 LD cycles did not show any increase in nitrogen fixation rates even when incubated under anaerobic conditions (Fig. S12).

**The Δ*kaiA* mutant exhibits disruption in cellular oxygen dynamics**
Since the significant reduction in nitrogenase activity in the mutant grown under CL or 20:4 LD cycles could be restored when an anaerobic incubation environment was provided, we hypothesized that the cellular oxygen dynamics were different between the WT and the Δ*kaiA* mutant under these extended light periods. To test this hypothesis we monitored the dissolved oxygen (DO) levels of cultures grown under CL as well as cultures subjected to CL after entrainment under 12:12 h LD cycles. Intracellular oxygen levels cannot be measured and as such, studies that attempt to assess the photosynthetic and respiratory activities of cyanobacterial cells monitor dynamic changes in the DO levels of the culture medium[26,27,35–37]. In a closed culture system, the DO levels in the media fluctuate, increasing with oxygen evolution during the photosynthetic phase and decreasing with oxygen uptake during respiration. Under 12 h LD conditions both in the WT and the mutant the oxygen level showed a sharp decline at the beginning of the dark period and began to rise at the end of the dark period (Fig. 4 a,b). When the entrained cells were subjected to CL, the WT maintained the sharp oscillations in the oxygen levels that coincided with the oscillations observed in OD 730, with the rise in oxygen coinciding with the rise in OD and vice versa (Fig. 4a, green arrows). However, in Δ*kaiA* these oscillations rapidly dampened upon transition to CL and they did not strictly coincide with the oscillations in OD 730 (Fig. 4b, orange arrows). When grown under CL without any prior entrainment, the WT culture exhibited sharp oscillations in oxygen levels (Fig. 4c). In contrast, in the Δ*kaiA* mutant, little and random dips in oxygen levels were observed (Fig. 4d,). In the WT nitrogenase activity could be detected in samples collected after the OD and oxygen levels began to dip (between D0 and D2, see methods) and at these time points quantum

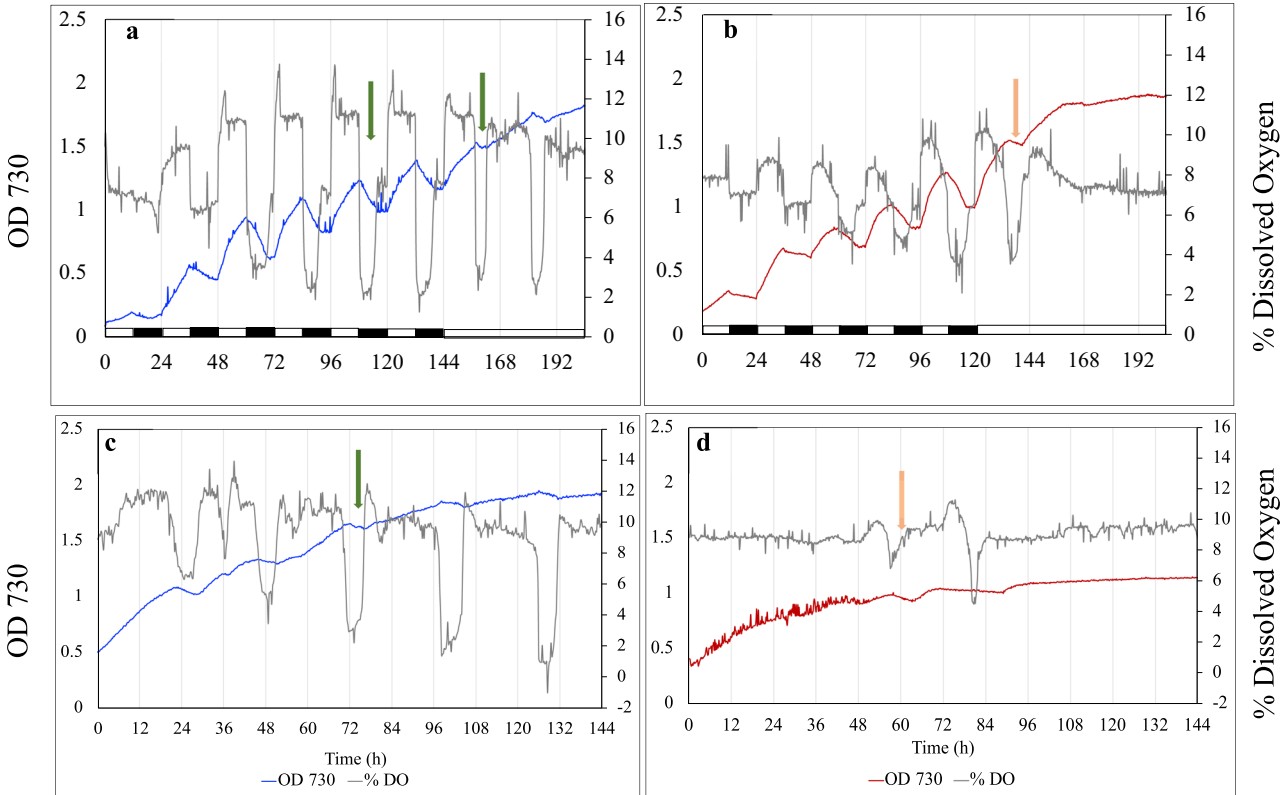

**Fig. 4 | Cellular oxygen dynamics in the WT and Δ*kaiA* strains.** Rhythms in growth and DO levels coincide with external LD cycles (green arrow) in the WT (**a**) and Δ*kaiA* (**b**). Rhythms are sustained in the WT after cells are transitioned to CL (green arrow) but not in the mutant (orange arrow). When grown under CL the WT (**c**) exhibits distinct endogenous rhythms in growth which coincide with oxygen cycling (green arrow). The rhythms are lost in Δ*kaiA* (**d**) where only a few random drops in oxygen levels are seen (orange arrow). Representative data from multiple runs are presented. 3 independent datasets for growth under CL (showing variability) for the WT and mutant have been provided as raw data (Source Data).

yields reflecting photosynthetic efficiency were low. In contrast, quantum yields were higher in samples that were collected after the OD and oxygen levels began to rise (between L0 and L2, see methods) and no nitrogenase activity could be detected in these samples. In contrast, only a weak correlation between nitrogenase and photosynthetic activities could be observed in the mutant at time points where drops and rises in OD and oxygen levels were discernable.

Analysis of DO level data from multiple runs revealed larger variations in absolute values among replicates of WT and Δ*kaiA*. However, we observed very similar patterns of periodicity in the different WT and Δ*kaiA* runs. For example, the WT had significantly larger fluctuations in the DO levels, and exhibited a clear oscillatory pattern (peaks and crests at constant intervals). In contrast, in the mutant, except for some random fluctuations observed during the experiment, DO levels remained mostly constant without any obvious oscillatory pattern (source data). We selected a representative sample from the WT and Δ*kaiA*, with a mean DO% of around 9%, for quantitative analysis of the DO levels over the experiment. Our analysis revealed that DO% in the WT sample fluctuated in the range of 5.2% to 8.8% within an 8 h period on consecutive days of the experiment (Supplementary Fig. 13). However, for the Δ*kaiA* mutant, we observed significant changes in DO% levels during only two windows in the entire duration of the experiment (2.5% and 6.1% on day3 and day4 respectively) and the DO% changed by less than 1% during the remaining days. As noted above, even though absolute values had larger variations among replicates, similar behaviors were observed in other samples of WT and the Δ*kaiA* mutants.

We assessed the photosynthetic and respiratory activities of cells fixing nitrogen in incubation bottles under aerobic incubation conditions. At the time point where the WT showed high rates of respiratory oxygen uptake (T6), the mutant did not, which instead exhibited higher photosynthetic oxygen evolution, indicating inability of the cells to shut off photosynthesis and initiate high rates of respiration during nitrogen fixation (Supplementary Fig. 14). Genes involved in photosynthesis showed no significant difference in expression in samples collected from the light phase (CL or L2 of a light dark cycle), but interestingly were significantly upregulated in the dark (D2) samples (Supplementary Fig. 11c). Together, these results indicate that the switch controlling the shutdown of the photosynthetic phase and onset of the respiratory phase is malfunctional in the Δ*kaiA* mutant, thereby disrupting its cellular oxygen dynamics.

## Instability of the nitrogenase enzyme in Δ*kaiA*

To determine if the altered cellular oxygen dynamics in the mutant affected the nitrogenase enzyme and its activity, we performed a western blot analysis of WT and Δ*kaiA* samples undergoing aerobic and anaerobic incubation for nitrogen fixation. During incubation in sealed vials, nitrogen fixation rates continue to increase for upto more than 48 hours and start to plateau around 60 hours. We sampled cells of the WT and the mutant 24 hours after the start of aerobic or anaerobic incubation and performed a western blot with the total cellular protein. We used the anti-nifD antibody to probe the molybdenum-iron proteins of the nitrogenase enzyme complex and compared the protein levels between the different samples. Since the antibody showed multiple bands, we also expressed the NifD protein in E.*coli* and used the purified protein as a positive control. A comparison of the equal amount of total cellular protein between the WT and mutant revealed a strong NifD band in the WT under aerobic incubation conditions (Fig. 5). In comparison, the mutant showed a very weak band, indicating reduced protein levels in the cells. In samples

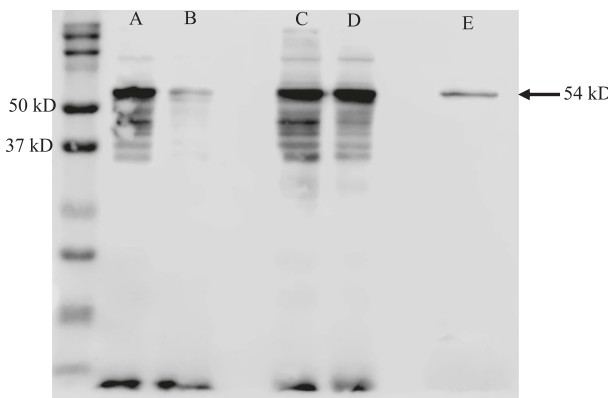

**Fig. 5 | Western blot analysis of the NifD protein in WT (A, C) and Δ*kaiA* (B, D) cells grown under CL and incubated for nitrogen fixation assay in air (A, B) or argon (C, D).** Total cellular protein was probed with NifD antibody. Experiments were independently repeated three times with similar results. *Cyanothece* 51142 NifD protein was expressed in *E.coli* and purified protein was used as a positive control (E). Arrow showing 54 kD NifD specific band in the samples. The image is a superimposition of the 700 channel on the chemiluminescence channel recorded on a Li-Cor ImageQuant LAS-4000 imager. The Bio-Rad Precision Plus Dual Color Protein Standard was used as the molecular weight marker.

incubated under anaerobic conditions, the NifD protein levels were similar in the WT and the mutant. These findings together with our observation that nitrogenase activity in the Δ*kaiA* mutant is revived under anaerobic incubation conditions suggest that unfavorable oxygen levels in the cells grown under CL or long day cycles lead to degradation of the nitrogenase enzyme and lower nitrogen fixation rates.

## Discussion

Since the discovery of nitrogen fixation in unicellular cyanobacteria, various studies have implicated a circadian clock function in segregating this oxygen-sensitive process from photosynthesis. However, a direct involvement of the clock in orchestrating this segregation could not be demonstrated. In this first-of-its-kind study, we investigate the function of the circadian clock in a unicellular diazotrophic cyanobacteria, specifically probing the role of KaiA, the protein that was the latest addition to the prokaryotic circadian clock, supposedly conferring the most advanced features to it[9,38]. Compared to KaiB and C, KaiA has undergone major structural modifications likely induced by selective pressures from a changing environment over the course of cyanobacterial evolution, suggesting that it played a key role in shaping the functions of the clock[39–41].

WT *Cyanothece* cells exhibit prominent oscillations in optical density (~24 hour period) when grown under CL and diazotrophic conditions, implying a strong circadian regulation of growth (Supplementary Fig. 3, Figs. 1, 2). A strong endogenous rhythm is also reflected in the robust oscillations in DO levels observed under these conditions (Fig. 4c). Such conspicuous rhythms in physiological processes, driven by the internal clock are not evident in strains currently used as models to understand the circadian function[42–44]. Therefore, *Cyanothece* 51142 offers a much-awaited opportunity to directly explore the effects of the clock on the physiology of an organism.

Since direct measurement of intracellular oxygen levels is not possible in cyanobacterial cells, changes in the DO levels of the culture are considered as a reflection of the changes in intracellular oxygen levels brought about by photosynthetic and respiratory activities of the cells[26,27,35–37]. Under CL and nitrogen-deficient conditions, the rhythmic rise and drop in OD and DO levels in the WT coincide and correspond with the photosynthetic and nitrogen fixation phases of the cells when oxygen evolution and respiratory oxygen uptake

predominate respectively. The precise peaks and troughs in the DO levels in the absence of an external diel cue suggest the involvement of tight internal switches that control the rise and drop in oxygen levels at the cusp of nitrogen fixation and photosynthetic activities, thereby segregating the incompatible processes. Earlier studies have documented the importance of the rhythms in oxygen cycling in maintaining high rates of aerobic nitrogen fixation in unicellular cyanobacteria[26,27]. These oscillations denoting the segregation of the photosynthetic and respiratory phases of the cells are lost in the Δ*kaiA* mutant, and instead, a few random dips in DO levels are seen (Fig. 4). qRT-PCR analysis revealed enhanced transcript levels of genes involved in respiration in the mutant (Supplementary Fig. 11b). The mutant also exhibited respiratory oxygen uptake (Supplementary Fig. 14b), suggesting that there are no intrinsic defects in its respiratory machinery and the disruption in oxygen cycling can most likely be attributed to a dysfunctional regulatory switch that stops photosynthesis and initiates high rates of respiration following metabolic cues. Along similar lines, a switch was shown to be involved in shutting off photosystem II when *Crocosphaera Watsonii* cells were required to fix nitrogen under light[29]. A switch was also implicated in a study in *Cyanothece* BG 043511, where regardless of the photoperiod, photosynthesis was turned off and nitrogen fixation was initiated 20 h after the onset of light[45].

The metabolic state of the cell has been suggested to be a factor responsible for entraining the phases of the internal clock in various systems[46]. In *Cyanothece* 51142, the availability of photosynthetically fixed carbon could possibly be a trigger for nitrogen fixation and depletion of the carbohydrate reserve along with the accumulation of cyanophycin, a nitrogen reserve, could be a cue for photosynthesis to start in the absence of external LD cues. Although the observed oscillations in *Cyanothece* may be triggered by metabolic cues, a role for KaiA in coordinating the processes is evident from the fact that the rhythms are abolished in the Δ*kaiA* mutant and are restored to WT levels upon re-introducing the *kaiA* gene.

Under CL, nitrogenase activity in the WT is lower compared to growth under 12:12 or 16:8 h LD cycles (Fig. 3d), and under this condition, the growth dynamics reflect phase separation of 16:8 h for photosynthesis and nitrogen fixation (rise and drop in OD) respectively, a rhythm the WT cells automatically resort to (Fig. 1a, S3). This reflects the efficiency of the clock in segregating photosynthesis and nitrogen fixation so that an optimal C/N ratio conducive to robust growth can be maintained. Under light/dark cycles, on the other hand, the onset and cessation of photosynthesis is regulated strictly by the presence and absence of light signals[47]. At the end of the light phase, the termination of photosynthesis is a cue for the cells to prepare for nitrogen fixation. As a first step towards such preparation, cyanobacterial cells ramp up the rates of respiration utilizing glycogen granules that are photosynthetically synthesized during the day. Such high rates of respiration are responsible for scavenging the intracellular oxygen, preparing the cells for nitrogenase activity[26,27]. Thus, under 12:12 and 16:8 h LD growth conditions the rhythms in photosynthesis and nitrogen fixation are imposed by the external LD cycles (Fig. 2a,b), thereby enforcing segregation of the processes, however in the absence of the day/night rhythm imposed by nature or under altered day length conditions, the internal clock function becomes critical in coordinating the processes.

While nitrogen fixation rates are higher in the WT under 12:12 and 16:8 h LD conditions, no advantage in biomass accumulation is seen (Fig. 3a, b, d, S7), indicating that the excess nitrogen is likely processed for storage, a necessity for cells to function under diurnal cycles but not under CL (Figs. 3a,b, 1b). The dampened or loss in oscillations of the Δ*kaiA* mutant observed under CL or when LD-grown cultures were subjected to CL correlated with reduced nitrogenase activity. The activity could be revived and rates higher than the WT could be attained when cells were subjected to an anaerobic incubation

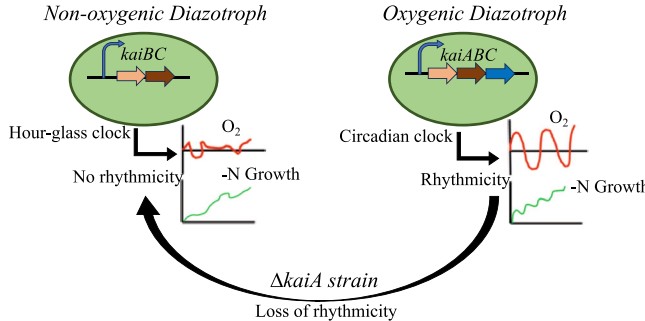

**Fig. 6 | Schematics showing the importance of KaiA in regulating cellular oxygen dynamics in unicellular diazotrophic cyanobacteria.** KaiA is essential for maintaining strong oscillations in oxygen levels which in turn is crucial for nitrogenase function. In the absence of KaiA the clock functions like a prokaryotic hour glass clock (*kaiBC* clock) and exhibits little rhythmicity.

environment. This suggests that the switch regulating the oscillations and the segregation of processes is controlled by KaiA and in the absence of this clock protein, ineffective segregation of photosynthesis from the nitrogen fixation phase of the cells results in suboptimal intracellular oxygen levels, causing damage to the nitrogenase enzyme and reduction in nitrogenase activity. This was further confirmed by our comparative analysis of the nitrogenase enzyme (NifD) levels in the WT and the mutant under aerobic and anaerobic incubation conditions (Fig. 5).

In *Synechococcus* PCC 7942 it has been shown that inactivation of *kaiA* leads to damped, low-amplitude oscillations in gene transcription and the *kaiBC* promoter activity has been suggested to be responsible for retaining the basal level of oscillations[48]. Since we were unable to delete *kaiC1* from *Cyanothece* 51142, we probed the effect of simultaneous deletion of *kaiA* and *B1* on the clock function by generating a Δ*kaiAB1* double mutant. The Δ*kaiAB1* mutant resembles the phenotype of a Δ*kaiA* mutant with a more stringent reduction in the amplitude of oscillations (Supplementary Fig. 15a, b), Further analysis of the double mutant might provide clues for identifying the source of the residual rhythms observed in the Δ*kaiA* mutant. Overall, these studies suggest a specific role for the KaiA-mediated circadian clock in timing physiological processes in unicellular diazotrophic cyanobacteria that perform aerobic nitrogen fixation so that an optimal intracellular level of oxygen conducive to sustained nitrogenase activity can be maintained.

Evolutionary studies imply an intricate link between the origin of *kaiA* and the evolution of aerobic diazotrophic cyanobacteria[9,49]. Evidence based on phylogenetic analysis suggests that cyanobacteria first emerged as anaerobic diazotrophs more than 3 billion years ago and subsequently evolved into microaerobic diazotrophs that relied on temporal separation of photosynthesis and nitrogen fixation[50,51]. An hourglass-type clock encoded by the *kaiBC* genes would have been a sufficient feature in anaerobic diazotrophs where protection of the oxygen-sensitive nitrogenase enzyme was inconsequential (Fig. 6, S16). Such an hourglass-type clock that does not exhibit sustained rhythms in the absence of the entraining cue, but is essential for maintaining daily rhythmicity in nitrogenase activity, is observed in extant anaerobic nitrogen-fixing strains like *Rhodobacter sphaeroides* and *Rhodopseudomonas plaustris*[52,53]. Some unicellular cyanobacteria also harbour the *kaiBC* operon suggesting that an hourglass-type clock could have been the time-keeping machinery in early cyanobacteria. KaiA has been suggested to be a cyanobacterial invention, possibly an addition to the hourglass type clock to initiate robust, sustainable rhythms in physiological processes (Fig. 6). This clock feature enabled cells to segregate oxygen-sensitive processes as they strived to adapt to increasing oxygen levels in the environment[53]. Interestingly, the evolution of the KaiA protein in unicellular diazotrophic cyanobacteria

has been linked to the oxygenation of the Metaproterozoic ocean around 1570–1600 Mya[38]. Alternatively, with the exception of G. *violaceus* PCC 7421 (which lacks all *kai* genes), KaiA is found in all extant cyanobacteria, including the phylogenetically ancient unicellular anaerobic nitrogen-fixing strains *Synechococcus* sp. JA-2-3B′a and JA-3-3Ab, implying an earlier time of *kaiA* origin[38,54]. While most strains have retained the gene, suggesting that it may confer an advantage in the present environmental scenario, some phylogenetically advanced ones indicate its likely dispensability. For example, *Prochlorococcus*, a non-diazotrophic marine cyanobacterial species that has undergone massive genome streamlining, harbors an hourglass type clock or possesses a truncated *kaiA*, indicating erosion of this gene from its genome[38,55]. Similarly, UCYN-A, a nitrogen-fixing cyanobacterial endosymbiont that has undergone genome reduction and lost its photosynthetic ability, only harbors the central oscillator KaiC[56,57]. These examples suggest that loss of the KaiA component of the clock may confer some advantages for strains that do not need to maintain robust self-sustaining oscillations. Such an advantage is perhaps reflected in the higher nitrogen fixation rates of the Δ*kaiA* mutant under anaerobic incubation conditions, likely due to the relaxation of unknown metabolic regulations that the KaiA-mediated clock imposes. The conditional dispensability of the gene is reflected in the fact that there is no significant growth disadvantage in its absence under nitrogen-sufficient conditions (Supplementary Fig. 17).

In conclusion, our findings demonstrate a prominent role for the KaiA-mediated circadian clock in regulating the oxygen levels in unicellular diazotrophic cyanobacteria, a function that is critical for their survival under aerobic, nitrogen-limited conditions. This regulation is likely mediated by a switch that responds to cellular metabolic cues and rapidly shuts off photosynthesis while simultaneously initiating high rates of respiration to scavenge intracellular oxygen, priming the cells for nitrogen fixation. Unlike *Synechococcus* PCC 7942, where the regulatory components of the cyanobacterial clock are well understood[32,48,58], the regulatory features of the circadian system are completely unknown in unicellular diazotrophic cyanobacteria. A preliminary analysis of some of the key regulatory players, including *rpaA*, the master regulator of the circadian clock, revealed differential regulation in the Δ*kaiA* mutant of *Cyanothece* 51142 (Supplementary Fig. 18). Elucidation of these regulatory pathways and their components will reveal unknown facets of circadian regulation, impacting research in various sectors including agriculture and health.

## Methods
### Culture conditions and media
WT *Cyanothece* ATCC 51142 and the *kai* mutants were cultured in ASP2 medium[59] in shaking flasks (150 rpm) at 30 °C under 100 μmol photons m$^{-2}$ s$^{-1}$ white light and ambient $CO_2$. WT and mutant strains were maintained on BG-11 agar plates under 100 μmol photons m$^{-2}$ s$^{-1}$ white light and ambient $CO_2$. For growth assays under nitrogen-fixing conditions and nitrogenase activity determination, WT and mutant cells were precultured in ASP2 medium with reduced nitrogen content (1.7 mM $NO_3^-$) for 3 days under 100 μmol photons m$^{-2}$ s$^{-1}$ white light and ambient $CO_2$. *E. coli* strains used in conjugation experiments were grown in LB medium with 200 rpm shaking at 37 °C.

### Generation of clock gene mutants
To generate markerless, Δ*kaiA*, Δ*kaiB1*, and Δ*kaiC1* mutants of *Cyanothece* 51142, we used the previously described CRISPR-Cpf1 technique[60]. Briefly, we used the pSL2680 vector as the base plasmid into which the gRNA sequences targeting the three *kai* genes were cloned to generate pSL2680-gRNA-*kaiA*, pSL2680-gRNA-*kaiB1*, and pSL2680-gRNA-*kaiC1*. Upstream and downstream regions (~700 bp) of the three genes, that would serve as the repair templates, were then cloned into the above plasmids by Gibson Assembly[61]. The accuracy of the sequences cloned were confirmed by sequencing (Genewiz®,

South Plainfield, NJ). The plasmids were maintained in *E. coli* strain XL1-Blue. The resultant plasmids were conjugated into WT *Cyanothece* 51142 following the method previously described[62]. Complete absence of the *kaiA* and *kaiB1* genes from the genome of *Cyanothece* 51142 could be verified by PCR analysis (Supplementary Fig. 2). Initially 6 completely segregated colonies were tested for growth and five of those six clones exhibited consistency in the observed phenotypes. Two of these clones (#1 and #3) were included in our studies. In contrast to Δ*kaiA* and *B1*, our efforts to generate a *kaiC1* mutant failed. Even after multiple rounds of streaking, the WT *kaiC1* gene persisted in the genome leading to non-segregated lines of this mutant. Therefore, we took a more conventional approach and aimed to replace the *kaiC1* gene with a kanamycin cassette by homologous replacement. The kanamycin-resistant colonies obtained after conjugation of this plasmid into the WT strain were subjected to very high antibiotic pressure (500 μg/mL) in liquid culture as well as on plates. However, we failed to obtain a completely segregated mutant. Similarly, our attempts to delete the entire *kaiAB1C1* cassette using CRISPR (two sets of gRNA's targeting *kaiA* and *kaiC1* were used) and conventional techniques failed. However, we were able to generate a Δ*kaiAB1* strain using CRISPR with pSL2680 vector as the base plasmid into which gRNA targeting the *kaiA* gene was introduced and 700 bp upstream and downstream of the *kaiA* and *kaiB1* genes respectively were used as repair templates. Three colonies from the Δ*kaiAB1* mutant were tested and exhibited consistent phenotypes. We also generated a Δ*kaiA'* complementation strain by introducing the WT *kaiA* gene into the deletion strain using CRISPR. Three colonies were tested for insertion of the *kaiA* gene and all three colonies were positive and showed similar growth patterns as the WT. Primers used for creating the various clock mutants as well as the complementation strain have been listed in Supplementary Table 2.

## Growth assays

For photoautotrophic growth, cells were added to fresh ASP2 – N media in glass tubes to a final volume of 60 ml with the optical density at 730 nm (OD730) between 0.1 and 0.2. Growth was continuously monitored for a period of 4–7 days (depending on the LD cycle used) using the MC1000 Multicultivator (Photon Systems Instruments, Czechia). The MC 1000 was set to maintain temperature at 37 °C and light intensity of 850 μmol photons m$^{-2}$ s$^{-1}$ during the growth period. Since under CL greater variability in oscillations was observed compared to LD conditions, more independent runs were performed under CL (raw data has been provided for 3 independent runs of the WT and 5 independent runs of the mutant as source data). After 96 h (CL and 12hLD growth) or 144 h (all LD to CL growth experiments) of growth 5 mL of culture of the WT and Δ*kaiA* was harvested from each growth condition to quantify biomass. Cells were centrifuged at 5000× *g* and washed twice with sterile H$_2$O. Whatman 0.7 μM GF/B glass microfiber filters (70 mm diameter) were weighed to obtain the filter weight. The washed cells were then added to the filter, and left to dry for 24 h at 60 °C. The filters with the dried cell mass were then weighed three times and the average weight of biomass was calculated. Single-factor ANOVA test was used to compare the total biomass accumulation of the two strains under different growth conditions, $p < 0.05$ being considered statistically significant.

## Mathematical approach for discerning rhythmicity

In order to mathematically model the growth data, we first removed small fluctuations in the measurements using a moving-average filter. We then employed exponential curve fitting to approximate the overall growth, which appears to saturate over time. Exponential curve can be represented using the equation "$y = a - b * \exp(-c * x)$", where y is the growth measurement, and x is the time. The parameters "a" captures the maximum (saturating) growth level, "b" represents the difference between final and initial growth measurements and "c

captures how fast samples reach the saturation level. The above curve was fitted to each of the samples for both WT and Δ*kaiA* and the respective parameters were determined.

To enhance the visibility of the cyclic patterns of the growth data as well as to perform quantitative analysis, we employed signal detrending. For this purpose, we subtracted the overall trend in the growth data from the actual measurements. The resulting detrended signal was analyzed using power spectrum density analysis. In order to identify the differences in the growth rates and the cyclic patterns of the growth data, we employed the Mann-Whitney U test to compare multiple samples of the WT ($n = 3$) and the Δ*kaiA* mutants ($n = 5$).

## Whole-cell absorption and pigment quantification

Cells growing under CL and LD conditions in multicultivators were sampled on day 2 (for CL cultures) or day 3 (for LD culture) and whole-cell absorption was measured using DW2000 (Olis, Inc., USA) in disposable plastic cuvettes with 1 cm pathlength. The whole-cell phycocyanin and chlorophyll contents were calculated from the absorption spectra using the formulae obtained from Arnon et al.[63].

## Nitrogenase, Photosynthetic, and Respiratory Activity Measurements

To determine the rates of nitrogen fixation 20 mL of culture (from a 2-d-old, CL or 3-d-old 12-h light/dark culture) was sampled at the beginning (at the onset of dark, D0 or 2 h into the dark period, D2) of the dark period (LD culture) or at the point when the OD 730 begins to drop (CL culture) from multi cultivators and transferred to air-tight glass vials (125 mL) and incubated in air under a light intensity of 100 μmol photons m$^{-2}$ s$^{-1}$ for 12 h. For anaerobic incubation, the glass vials were flushed with argon for 5 – 10 min. Nitrogenase activity of the cultures was determined using an acetylene reduction assay[64] and expressed in terms of the ethylene produced[27]. Ethylene that accumulated in the head space of sealed culture vials was withdrawn with an air-tight syringe and quantified using an Agilent 6890 N gas chromatograph equipped with a Poropak N column and a flame ionization detector using argon as the carrier gas (flow rate of 65 ml min$^{-1}$), following the manufacturer's instructions[27]. The temperature of the injector, detector, and oven were 150, 200 and 100 °C, respectively. Total chlorophyll *a* was extracted by methanol and quantified spectrophotometrically using an Olis DW2000 spectrophotometer (OnLine Instrument Systems, Inc., GA). At least 3 biological replicates and 5 technical replicates were included for the WT and the Δ*kaiA* strains under each condition. Single-factor ANOVA test was used to compare the total biomass accumulation of the two strains under different growth conditions, $p < 0.05$ being considered statistically significant.

The respiratory activities of the cells were measured in photobioreactors equipped with an integrated Mettler-Toledo Clark-type oxygen electrode[27]. The vented bioreactors were operated in stirring mode to evaluate the changes in dissolved oxygen concentration of the cultures from photosynthetic and respiratory activities of the cells. The cultures were transferred to the bioreactors at the beginning of the log phase of growth and studied in batch culture mode under stirring conditions for 6 days. Similar to our growth experiments, under CL greater variability was observed in the DO levels compared to LD growth conditions. Raw data for 3 representative runs of the WT and the mutant have been provided as a source data file. Photosynthetic oxygen evolution and respiratory oxygen uptake of cells were measured in samples from the incubation vials using a Clark-type electrode[27,65] at the beginning of the incubation period (T0) and 6 h after incubation (T6). Cyanobacterial cells were normalized on the basis of chlorophyll content to determine the quantum efficiency of the WT and mutant strains using the FL-200 dual modulation PAM fluorometer. To compare the changes in DO levels, we computed the

mean DO% levels as well as the absolute changes in the DO% over the period.

## Analysis of gene transcripts

50 ml of WT and Δ*kaiA* cells grown in multi cultivators under nitrogen fixing, CL or LD conditions were harvested at the cusp of the photosynthetic phase and nitrogen fixation phase (the point at which OD 730 begins to drop, figure) (if CL culture) or at L2 and D2 time points (if LD culture). Cells were either prepared directly for RNA isolation or subjected to incubation under aerobic or anaerobic conditions in air-tight bottles and harvested after 1 h. Total RNA from each sample was isolated using RNAwiz reagent (Ambion, Inc.) according to the manufacturer's instructions and treated with RNase-free DNase I prior to cDNA synthesis using random primers (Invitrogen) and Superscript II reverse transcriptase.

qRT-PCR was performed in a CFX96 real-time PCR system (Bio-Rad), following the protocol described in[66]. A minimum of three replicates were performed for each of the samples. All primers used for qRT-PCR are listed in Supplementary Table 3. No-template reactions were used as negative controls. ΔCt was calculated by subtracting the Ct of the endogenous reference gene (*16 s rRNA*) from the Ct of the genes under investigation. The relative changes in transcript levels of the genes were analyzed using the $2^{-\Delta\Delta CT}$ method. To calculate ΔΔCt, the WT ΔCt was subtracted from the ΔCt of Δ*kaiA*. The fold change in expression of the Δ*kaiA* genes was calculated according to[67].

## Western blot analysis

WT and Δ*kaiA* cells (60 ml culture) grown under CL and nitrogen-fixing conditions were harvested at the cusp of the photosynthetic phase and nitrogen fixation phase (the point at which OD 730 begins to drop) and subjected to incubation under air and argon in air-tight bottles. Cells were harvested after 24 h of incubation in the bottles, after confirming nitrogenase activity in parallel experiments. Cells were washed and the pellet resuspended in 0.5 ml TG buffer (10 mM Tris-HCl [pH 8.0], 10% glycerol) containing a protease inhibitor cocktail (Sigma-Aldrich) and DNAse I. 0.5-ml volume of sterilized, cold, acid-washed glass beads was added to the cells, and the cells were disrupted by vortexing (BioSpec Products). The mixture was allowed to settle and the supernatant was collected for centrifugation at 3000 × *g*, for 5 mins. The supernatant was collected and further subjected to 30 mins of centrifugation at 16000 X g. The final supernatant was transferred into a new tube and the total soluble protein was quantified using bicinchoninic acid (BCA) protein assay reagent (Thermo Scientific).

10-μg (total) protein extract from each sample was separated on a sodium dodecyl sulfate-polyacrylamide (12.5% [wt/vol]) gel by electrophoresis. The NifD protein, expressed and purified as previously described[68], was loaded at a concentration of 2 μg. Separated proteins were transferred to a polyvinylidene difluoride (PVDF) membrane (Millipore) for western blot analysis. The anti-nifD antibody[69,70] was used at 1:2,000 dilution. Bands were visualized using chemiluminescence reagents (Millipore-Sigma) with a LI-COR Odyssey Fc (LI-COR Biotechnology) imager. 3 blots were generated from independent biological samples (3 each of WT and mutant) and a representative blot is presented in Fig. 5.

## Reporting summary

Further information on research design is available in the Nature Portfolio Reporting Summary linked to this article.

# Data availability

All data supporting the findings of this study are provided within the article and the supplementary files. All further correspondence, additional information and material requests should be addressed to the corresponding author. Source data are provided with this paper Source data are provided with this paper.

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

## Acknowledgements

This study was supported by the National Science Foundation (NSF) Grant MCB 1933660 to H.B.P. We would like to thank all Pakrasi Lab members for collegial discussions. We would also like to acknowledge our lab technician Xinjun Duan for her help with day-to-day activities that supported this study.

## Author contributions

H.B.P. conceptualized this study. A.B. and H.B.P. designed the experiments. A.B. and A.S. conducted experiments and A.B., A.S., and T.E. analyzed data. A.B. and H.B.P. wrote the manuscript. All authors reviewed and revised the manuscript.

## Competing interests

The authors declare no competing interests.
