## [Peer Review File · Nature Communications]

Endogenous clock-mediated regulation of intracellular oxygen dynamics is essential for diazotrophic growth of unicellular cyanobacteriaReviewer #1 (Remarks to the Author):

NCOMMS-23-08415

Review report

6 April 2023

Nitrogenase, the enzyme responsible for the nitrogen fixation reaction, is very vulnerable to oxygen. Most nitrogen-fixing organisms that inhabit under environments containing oxygen develop a variety of protection mechanisms of nitrogenase from oxygen. For nitrogen-fixing cyanobacteria, the protection of nitrogenase is a much harder task than other heterotrophic organisms since they produce oxygen by photosynthesis. Some nitrogen-fixing cyanobacteria, such as *Anabaena*, solve this problem by developing heterocysts that are specialized for nitrogen fixation to segregate spatially from photosynthesis in vegetative cells. On the other hand, some cyanobacteria that do not differentiate heterocysts are known to solve this problem by temporal separation of these incompatible processes in the same cells.

The observation of temporal separation of nitrogen fixation and photosynthesis in nitrogen-fixing cyanobacteria with a 24-hour cycle (Mitsui et al. 1986; Grobbelaar et al. 1986) provided the first experimental evidence for the presence of a robust circadian biological clock even in prokaryotes, which was subsequently confirmed by the discovery of the clock genes *kaiABC*. However, detailed genetic and biochemical analyses of the *kaiABC* genes and the KaiABC protein complex have been conducted exclusively on the non-nitrogen-fixing unicellular species *Synechococcus* sp. PCC 7942 (*Synechococcus* 7942). Little research has been done on actual involvement of the KaiABC clock in the temporal separation of nitrogen fixation and photosynthesis in nitrogen-fixing cyanobacteria. In this paper, Bandyopadhyay, Sengupta and Pakrasi attempts to clarify this point using the unicellular nitrogen-fixing cyanobacterium *Cyanothece* sp ATCC 51142 (*Cyanothece* 51142).

The authors isolated a *kaiA*-knockout mutant Δ *kaiA* in *Cyanothece* 51142 and conducted the phenotype analysis of Δ *kaiA*, and concluded that KaiA is involved in the periodical drops in dissolved oxygen level supporting aerobic nitrogen fixing growth of this cyanobacterium. Studies in *Synechococcus* 7942 indicated that circadian rhythm generated by KaiABC regulates global expression of a large number of genes. Thus, the temporal separation of nitrogen fixation and photosynthesis in *Cyanothece* 51142 is also likely achieved through the global gene regulation by KaiABC. Although the authors claim a unique function for KaiA (regulation of oxygen dynamics) that is distinct from the function of circadian clock, it is not clear whether the observed phenotype is caused by an abnormal circadian clock due to the loss of KaiA or by a loss of a novel function of KaiA. To clarify this point, it is necessary to compare the Δ *kaiA* phenotype with that of another mutant in which the clock function is totally lost, i.e., a Δ *kaiC* mutant. Since KaiA has a regulatory role for KaiC activities (phosphorylation and ATPase activity), it is unlikely that KaiA directly contributes to the decrease in dissolved oxygen levels. Rather, the authors should clarify how the cyanobacterial cells decrease dissolved oxygen level in dark phase. In addition, although the title claims "KaiA regulates intracellular oxygen dynamics", what is being experimentally measured is only the dissolved oxygen level, the oxygen consumption rate and nitrogenase activity in wild type and Δ *kaiA*. In other words, the intracellular oxygen dynamics is only an estimate based on changes in these measurements. Thus, this reviewer has the impression that this expression is somewhat of an overstatement.

Transcription of *nif* genes and nitrogenase activity in nitrogen-fixing organisms including nitrogen-fixing cyanobacteria are regulated not only by oxygen levels but also by intracellular nitrogen status. Therefore, it is possible that the circadian periodicity of growth is caused just by cyclic changes in nitrogen status: nitrogen deficiency after photosynthetic growth during the day and nitrogen sufficiency by operation of nitrogen fixation at night. In other words, the cyanobacterial cells initially grow by photosynthesis, but after a certain time, nitrogen is depleted leading to induction of the *nif* genes and/or activation of nitrogenase activity, and then nitrogenase activity increases and nitrogen is supplied, which results in a state of nitrogen sufficiency and subsequent photosynthetic growth can be supported. It is possible that these cyclic changes of nitrogen status produce an apparent circadian rhythm in nitrogen fixation. This paper does not consider such a possibility at all.

The reviewer found a number of typos and points for improvement in the figures but considers

those remarks to be premature at this stage.

Reviewer #2 (Remarks to the Author):

The manuscript reports the exciting expansion of circadian analysis to the diazotrophic cyanobacterium *Cyanothece* ATCC 51142. So far the cyanobacteria are the only prokaryotes with a well-studied circadian clock, and most of the data come from a single species, *Synechococcus elongatus*, that does not fix nitrogen. The initial reports of cyanobacterial circadian rhythms were based on observations of striking alternations between photosynthesis and nitrogen fixation in some intractable unicellular strains, but mechanistic progress in understanding this cyclic physiology is sorely lacking. It is a major step forward to conduct genetic studies in *Cyanothece* ATCC 51142. The strength of the manuscript is in the demonstration that a *kaiA*-deletion strain, lacking an expected component of the circadian clock, has defects in oxygen dynamics that compromise nitrogen fixation under aerobic conditions. This potentially exciting finding is diminished by shortcomings in the manuscript.

Major concerns:

1. The genetic approaches leave questions unanswered about the specialness of KaiA for the observed phenotypes. The authors tried to inactivate each of the genes, *kaiA*, *kaiB*, and *kaiC* individually, and could not obtain *kaiC* mutants. However, they did not try to remove the entire 3-gene locus, which may have enabled *kaiC* to be deleted. The concern is that some residual rhythms remain, so do the observed phenotypes result from loss of a function of KaiA per se, or because a defective clock is doing something it shouldn't. Perhaps a *kaiA kaiB* double mutant would have been fully arrhythmic? However, it may very well be that *kaiC* can't be deleted and the *kaiA* deletion is the best test that can be obtained, which still allows valuable investigation. However, rigor in making sure that this mutant is as expected is missing. Were multiple independent clones tested, does a complemented mutant regain WT phenotypes, and/or was whole genome sequencing performed to make sure that deletion of *kaiA* did not select for second-site mutations? The first approach, and one or the other of the second and third, should be documented before drawing strong conclusions about the role of *kaiA* to make sure that the phenotypes result from its loss.
2. There is a general lack of rigor in the reported data. Individual traces are shown with no evidence of biological replicates, statistical analysis, or use of programs to discern rhythmicity. The phenotypes seem to have been "eyeballed," and in some cases the statements don't match the figures (such as regarding growth rate, when modest difference in biomass were found – without quantification – but "rates" were very similar for the first half of the experiments). Statements about phases are vague, as is the basis for the arrows in Fig 2.
3. The potentially exciting conclusion that the clock, or perhaps KaiA specifically, is involved in the oxygen protection of nitrogenase is suggested by these preliminary data, but not sufficiently supported to justify the evolutionary arguments presented.

Responses to the reviewers

Reviewer 1

The authors isolated a *kaiA*-knockout mutant Δ *kaiA* in *Cyanothece* 51142 and conducted the phenotype analysis of Δ *kaiA*, and concluded that KaiA is involved in the periodical drops in dissolved oxygen level supporting aerobic nitrogen fixing growth of this cyanobacterium.

Studies in *Synechococcus* 7942 indicated that circadian rhythm generated by KaiABC regulates global expression of a large number of genes. Thus, the temporal separation of nitrogen fixation and photosynthesis in *Cyanothece* 51142 is also likely achieved through the global gene regulation by KaiABC.

As stated in our manuscript, studies in *Synechococcus* 7942 have indeed demonstrated that the circadian rhythm generated by the *kaiABC* clock regulates global expression of a large number of genes and the reviewer is correct in pointing out that global gene regulation is likely to be involved in the temporal separation of photosynthesis and nitrogen fixation in *Cyanothece* 51142. RNA-seq studies are currently underway to investigate the changes in the global transcript profile of the cells in the absence of the KaiA component of the clock and identify players downstream of the clock that are involved in regulating key metabolic processes. In this study, we have performed RT-PCR analysis to capture changes in the transcript levels of certain groups of genes of interest and we also see significant changes at the transcript levels. In addition, we have investigated the transcription pattern of some of the regulatory genes that have been identified as key players in S7942 [supplementary figure (S18) added to the revised version of manuscript]. Some of these genes exhibit differential regulation in the Δ *kaiA* mutant with respect to the WT, while others do not. Thus, a global transcript profile of the mutants will be essential to understand the clock-mediated regulatory processes in unicellular diazotrophic cyanobacteria. The current manuscript introduces the idea that the clock can have a direct impact on the metabolism and physiology of an organism and an elegant system that allows easy scoring of such changes. This study pioneers the use of genetic tools in deciphering the clock function in a diazotrophic cyanobacterium and lays the foundation for further exciting discoveries in this field.

Although the authors claim a unique function for KaiA (regulation of oxygen dynamics) that is distinct from the function of circadian clock, it is not clear whether the observed phenotype is caused by an abnormal circadian clock due to the loss of KaiA or by a loss of a novel function of KaiA. To clarify this point, it is necessary to compare the Δ *kaiA* phenotype with that of another mutant in which the clock function is totally lost, i.e., a Δ *kaiC* mutant. Since KaiA has a regulatory role for KaiC activities (phosphorylation and ATPase activity), it is unlikely that KaiA directly contributes to the decrease in dissolved oxygen levels.

KaiA is an integral part of the cyanobacterial KaiABC clock and we did not intend to imply that its function is independent of the clock. Our study suggests that the KaiABC clock is no longer able to function in a normal way without KaiA. Our manuscript may have projected KaiA's function as independent of the clock in some ways and we thank the reviewer for pointing this out. We have modified our manuscript, including the title to emphasize that deletion of *kaiA* disrupts the overall clock function such that it behaves more like an hour glass clock (KaiBC clock), present in non-oxygenic diazotrophic microbes. Our study unequivocally demonstrates that the KaiA component of the circadian clock is responsible for the strong oscillations in growth and dissolved oxygen levels observed in the WT. A recently generated Δ *kaiA*' complementation strain is able to restore the oscillations to the WT levels, further affirming our conclusions (data added as a supplementary figure, Fig. S4). Along similar lines, a role for KaiA in enhancing and maintaining strong oscillations in transcript profiles of S7942 has also been reported (Kawamoto et al., 2020, cited in the revised manuscript – Ref48, Lines 474 to 476), further corroborating our findings that KaiA is responsible for enhancing the amplitude of the oscillations.

As stated in our manuscript, all our efforts (including our more recent efforts, also stated in the manuscript, Lines 113 to 120) to delete *kaiC* have failed, indicating that unlike S7942, this gene is essential for *Cyanotheca* and is likely responsible for the dampened rhythms maintained in the Δ *kaiA* mutant. In this revised manuscript we provide growth data for a Δ *kaiABI* double mutant that we recently generated and are currently investigating (Fig S15, lines 476 to 483). Although the Δ *kaiBI* mutant exhibits conspicuous oscillations distinct from the WT (Fig. S3), the Δ *kaiABI* mutant does not. Instead, it resembles a Δ *kaiA* mutant, suggesting that the dampening in oscillations results from a loss of *kaiA*.

Rather, the authors should clarify how the cyanobacterial cells decrease dissolved oxygen level in dark phase.

Under diurnal cycles, the onset and cessation of photosynthesis is regulated strictly by the presence and absence of light signals (Sicora et al. 2018, Ref47). At the end of the light phase, termination of photosynthesis is a cue for the cells to prepare for nitrogen fixation. As a first step towards such preparation, cyanobacterial cells ramp up the rates of respiration utilizing glycogen granules that are photosynthetically synthesized during the day. Such high rates of respiration are responsible for scavenging the intracellular oxygen. We have cited publications that clarify this point brought up by the reviewer (Ref 42, 43, 44, 26, 27, Lines 435 to 441). Our study shows that in the absence of the day / night rhythm imposed by nature, the internal clock function becomes essential in coordinating the processes.

In addition, although the title claims “KaiA regulates intracellular oxygen dynamics”, what is being experimentally measured is only the dissolved oxygen level, the oxygen consumption rate and nitrogenase activity in wild type and Δ *kaiA*. In other words, the intracellular oxygen dynamics is only an estimate based on changes in these measurements. Thus, this reviewer has the impression that this expression is somewhat of an overstatement.

The dissolved oxygen levels are a direct reflection of the cellular activities and the intracellular oxygen level of the cells. During photosynthesis phase of the cells, the DO levels increase and as photosynthesis shuts down and respiration takes over, the oxygen levels drop. Since the intracellular oxygen levels cannot be directly measured, the dissolved oxygen level is the best indicator of the intracellular levels and as such, several studies have used the DO levels to explain the cellular activities in *Cyanotheca*. We have ensured to include these citations to clarify the point (Ref 42, 43, 44, 26, 27, Lines 387-91).

Transcription of *nif* genes and nitrogenase activity in nitrogen-fixing organisms including nitrogen-fixing cyanobacteria are regulated not only by oxygen levels but also by intracellular nitrogen status. Therefore, it is possible that the circadian periodicity of growth is caused just by cyclic changes in nitrogen status: nitrogen deficiency after photosynthetic growth during the day and nitrogen sufficiency by operation of nitrogen fixation at night. In other words, the cyanobacterial cells initially grow by photosynthesis, but after a certain time, nitrogen is depleted leading to induction of the *nif* genes and/or activation of nitrogenase activity, and then nitrogenase activity increases and nitrogen is supplied, which results in a state of nitrogen sufficiency and subsequent photosynthetic growth can be supported. It is possible that these cyclic changes of nitrogen status produce an apparent circadian rhythm in nitrogen fixation. This paper does not consider such a possibility at all

As cited in our manuscript, a role for the circadian clock in segregating photosynthesis and nitrogen fixation was suggested many decades ago with the observation that unicellular cyanobacteria can fix nitrogen under continuous light. This study takes us a significant step further in understanding the role of the clock in such a segregation of incompatible processes. The reviewer also correctly points out that the metabolic demand of the cells can regulate the photosynthetic and nitrogen fixing activities and such intracellular metabolic signals can serve as a cue for the clock. We have included a brief discussion on the role of the metabolic status of the cell in coordinating intracellular processes (Lines 410-29). However, even if mediated by metabolic signals, the role of KaiA in the observed cyclic changes is unambiguous as

is evident from the fact that disrupting the clock by eliminating *kaiA* abolishes the oscillations in growth and oxygen cycling.

The reviewer found a number of typos and points for improvement in the figures but considers those remarks to be premature at this stage.

We have thoroughly gone through the manuscript to correct all typos and have revised some of the figures that we thought could have been confusing to the readers. Again, we thank the reviewer for the thoughtful suggestions. They definitely helped improve the manuscript.

Reviewer 2

The manuscript reports the exciting expansion of circadian analysis to the diazotrophic cyanobacterium *Cyanothece* ATCC 51142. So far the cyanobacteria are the only prokaryotes with a well-studied circadian clock, and most of the data come from a single species, *Synechococcus elongatus*, that does not fix nitrogen. The initial reports of cyanobacterial circadian rhythms were based on observations of striking alternations between photosynthesis and nitrogen fixation in some intractable unicellular strains, but mechanistic progress in understanding this cyclic physiology is sorely lacking. It is a major step forward to conduct genetic studies in *Cyanothece* ATCC 51142. The strength of the manuscript is in the demonstration that a *kaiA*-deletion strain, lacking an expected component of the circadian clock, has defects in oxygen dynamics that compromise nitrogen fixation under aerobic conditions. This potentially exciting finding is diminished by shortcomings in the manuscript.

We are very grateful to the reviewer for a thorough and critical analysis of our manuscript and for providing concrete suggestions that helped improve it significantly. We are also happy that the reviewer is in agreement with us that the findings are new and exciting and we have followed and implemented all the suggestions to overcome the shortcomings and the concerns that have been raised.

Major concerns:

1. The genetic approaches leave questions unanswered about the specialness of KaiA for the observed phenotypes. The authors tried to inactivate each of the genes, *kaiA*, *kaiB*, and *kaiC* individually, and could not obtain *kaiC* mutants. However, they did not try to remove the entire 3-gene locus, which may have enabled *kaiC* to be deleted. The concern is that some residual rhythms remain, so do the observed phenotypes result from loss of a function of KaiA per se, or because a defective clock is doing something it shouldn't.

Perhaps a *kaiA kaiB* double mutant would have been fully arrhythmic? However, it may very well be that *kaiC* can't be deleted and the *kaiA* deletion is the best test that can be obtained, which still allows valuable investigation. However, rigor in making sure that this mutant is as expected is missing. Were multiple independent clones tested, does a complemented mutant regain WT phenotypes, and/or was whole genome sequencing performed to make sure that deletion of *kaiA* did not select for second-site mutations? The first approach, and one or the other of the second and third, should be documented before drawing strong conclusions about the role of *kaiA* to make sure that the phenotypes result from its loss.

We thank the reviewer for the suggestions about confirming the correlation of the observed phenotypes with the absence of *kaiA*. Following the suggestions, in the past several months we have attempted to delete the entire *kaiABIC1* cluster from *Cyanothece* 51142 using CRISPR and conventional methods. However, we have remained unsuccessful in obtaining any segregated mutant colonies. However, following the reviewer's suggestion we have successfully generated a Δ *kaiAIB1* double mutant and performed assays to determine the effect of the double mutation on the strain phenotype (Fig S15). Although the Δ *kaiB1* mutant exhibits conspicuous oscillations distinct from the WT (Fig. S3), the Δ *kaiAB1* mutant does not and resembles a Δ *kaiA* mutant, suggesting that the dampening in oscillations

results from a loss of *kaiA* and that the residual rhythms observed in the mutant are likely regulated by *kaiC*. We did test multiple clones of the *kaiA* mutant and the information has been added in the manuscript (Lines 566-69, Supplementary file 1). Also, following the suggestions of the reviewer, we generated a complementation strain by introducing the *kaiA* gene into the $\Delta kaiA$ mutant (Fig S4, Lines 140-44). The complementation strain exhibits WT like traits in growth and nitrogen fixation confirming that the phenotypes observed are a direct effect of the deletion of the KaiA component of the clock. We have added our new findings as supplementary data (Fig. S4, S15). We have also cited an earlier study in S7942 which shows that loss of *kaiA* leads to dampened oscillations of transcript levels, thus supporting the role of KaiA in maintaining robust oscillations (Kawamoto et al., 2020, Ref48).

2. There is a general lack of rigor in the reported data. Individual traces are shown with no evidence of biological replicates, statistical analysis, or use of programs to discern rhythmicity. The phenotypes seem to have been “eyeballed,” and in some cases the statements don’t match the figures (such as regarding growth rate, when modest difference in biomass were found – without quantification – but “rates” were very similar for the first half of the experiments). Statements about phases are vague, as is the basis for the arrows in Fig 2.

We have provided multiple supplemental datasets in the revised version of the manuscript which include multiple biological replicates for the growth and oxygen cycling data (Supplementary file1 and 2). In the manuscript we had reported that there are some variability in the Δkai mutant and provided some of the variability observed in growth as supplemental data. These variability do not contradict our findings and in fact reinforces the importance of KaiA and the clock in general for regulating daily rhythms.

Eliminating the clock can not only dampen the rhythm but also generate abnormal rhythms and various phenotypic traits as has been reported in other systems.

We do realize that some of the statements in the manuscript were vague and not sufficiently supported by data. Taking into consideration the reviewer’s suggestion that mathematical validations of the results would make the study much more robust, we have brought in a new author, Thanura Elvitigala, who is an expert in mathematical modelling studies. He has used mathematical programs to discern rhythmicity in the observed phenotypes and our analysis has been added as supplemental material in the manuscript (Fig S5. S6. S13). We have also included biomass data in the manuscript and have made all comparisons using biomass data rather than growth (Fig. S7 and S9). Accurate growth rate estimation is not possible because of the oscillatory nature of the growth curves and therefore we have not included any growth rate comparison. We understand that the arrows in figure 2 were confusing and not adding much to our core conclusions in the manuscript and therefore we have removed them. All of our comparisons are based on the amplitude of the oscillations observed in the growth curve and actual biomass accumulated at the end of the run. We have also provided statistical analysis of the data wherever possible. All of the above changes have made our data more robust and have substantially improved the quality of the manuscript and we are very grateful to the reviewer for all the constructive criticism provided.

3. The potentially exciting conclusion that the clock, or perhaps KaiA specifically, is involved in the oxygen protection of nitrogenase is suggested by these preliminary data, but not sufficiently supported to justify the evolutionary arguments presented.

We agree with the reviewer that the evolutionary timeline in the model is not sufficiently supported by the findings in our current study and therefore we have revised the original model and included it as a supplementary figure (Fig S16). Several prior work that we have cited in the discussion, support our conclusion that in the absence of *kaiA* the clock functions like an hour glass clock and that *kaiA* is an addition to the *kaiBC* clock. We have included a simplified model (Fig 6) to highlight our main conclusions from this work: KaiA is essential for maintaining strong oscillations in oxygen levels under continuous light growth, which in turn is crucial for nitrogenase function. In absence of KaiA the clock functions like a prokaryotic hour glass clock which also maintains dampened rhythm.

We thank reviewer 2 for all of the above suggestions. We have conducted all suggested experiments, have performed extensive data analysis and modified the manuscript which has helped improve it substantially.

Reviewer #1 (Remarks to the Author):

Review report for NCOMMS-23-08415A
January 11, 2024

This paper is a revised version of the paper submitted in March. Bandyopadhyay et al. showed the importance of the circadian clock in nitrogen-fixing growth in the unicellular diazotrophic cyanobacterium *Cyanothece* sp. ATCC 51142, which has not been able to be analyzed in a molecular biological aspect due to the lack of gene manipulation technique. As the authors emphasized in the response letter, this paper demonstrated that the circadian clock is important for nitrogen fixing growth under aerobic conditions in *Cyanothece* sp. ATCC 51142 using the *kaiA*-disrupted mutant for the first time. This reviewer appreciates this point. However, Reviewer 1 still felt that the paper tended to overstate the experimental results in many places, which is included in the comments below:

Major comments

1. The title has been revised by the addition of "component of the circadian clock". However, this reviewer feels that this title itself is somewhat an exaggeration and would like to suggest the following title:

Oxygen dynamics produced by the circadian Kai clock is essential for maximal aerobic nitrogen-fixing growth in unicellular diazotrophic cyanobacterium *Cyanothece* sp. ATCC 51142

2. In Figure 3, the authors show that nitrogenase activity under anaerobic conditions is higher than under aerobic conditions, indicating that oxygen dynamics have a significant effect on nitrogenase activity. The acetylene reduction activity of nitrogenase is competitively inhibited by N₂. Therefore, the increase in activity on the argon sample compared to simply Air (i.e., 80% N₂) sample will include both an increase in activity with argon due to the loss of competitive inhibition by N₂ and an increase in activity with argon due to the relaxation of oxygen-induced inactivation of nitrogenase under anaerobic conditions. To clarify this, a comparison of activity with 20% oxygen-argon and 100% argon may be necessary.

3. Given that nitrogenase activity is higher under anaerobic conditions than WT, this reviewer would like to see nitrogen-fixing growth under anaerobic conditions in addition to the comparison of nitrogenase activity (Figure 3).

4. The subtitle of line 251 is overstated. How about the following?: "Instability of nitrogenase in Δ kaiA"

Minor points

5. Line 106, Please show the locus tags (cce_xxxxx) for *kaiA*, *kaiB1*, and *kaiC* genes.

6. Typo, *kaiB1* and *kaiC1* are written as *kaiB* and *kaiC*, respectively. (lines 109-117, 349, 355...)

7. Line 123-125, please clearly describe that growth is nitrogen-fixing growth (photosynthesis + nitrogen fixation). This important information is just shown as "in nitrogen deficient media" in the legend in Figure 1.

8. Description of the result of Figure 1 (lines 126-127), The authors simply describe "the WT exhibits an endogenous rhythm", but what is observed in the growth curve is not an endogenous rhythm, but rather a staircase-like growth. This reviewer feels the following expression should be better: Cells grow during subjective light period (increase in OD730) and do not grow during subjective dark period (constant or decrease in OD730), and these repetitions create an apparent endogenous rhythm in growth.

9. Fig. S8, panels C and D, Phycocyanin and chlorophyll contents are shown as Fold change (WT/ Δ kaiA), but should be Δ kaiA/WT if WT is the standard. In addition, no explanation is given for the color indication on the right sides of panels C and D.

10. Figure 3, Nitrogenase activity is shown as relative values, but the actual values should be indicated somewhere.

11. Figure 4, Vertical axis says OD720, but it should be OD730, as described in the text.

12. Figure S17, The description in the legend "nitrogen deficient media" does not match the description in the text, lines 391-392 "under nitrogen sufficient conditions". Also, the information on light conditions (when LD change to CL) is not stated.

14. Figure 5, In the Western blot analysis, only NifD was detected even though they used an anti-NifDK antibody. Please explain this point briefly. Also, the description regarding expression and purification of NifD protein that was used as a marker should be described in the Materials and

Methods section.

Reviewer #2 (Remarks to the Author):

The revised manuscript is greatly improved, and addresses most of the concerns of the reviewers. A few problems with presentation remain.

1. The title still suggests that there is a special role for KaiA in oxygen dynamics that is different from a role in completing the circadian clock mechanism. An accurate title would be: KaiA deletion in a unicellular diazotrophic cyanobacterium implicates the circadian clock in regulating intracellular oxygen dynamics

2. Lines 203-206:

When cells grown under 12:12h LD cycles were incubated under aerobic conditions, the average specific rates of nitrogenase activity in the Δ kaiA mutant was almost similar to that of the WT, whereas mutant cells grown under 16:8h LD cycles exhibited nitrogen fixation rates of up to 80 to 90 % that of the WT (Fig. 3a,b).

A few things about this wording are confusing. "almost similar to" doesn't make sense. They were almost identical, or they were similar? And this result is contrasted (whereas) with a condition that was 90% the same as WT. How is that not "almost similar"? Think about what point you want to make and be sure to say it clearly – do you want to show how findings are alike, or point out differences? Also, the lead in with "When cells grown...were incubated under aerobic conditions..." followed by "whereas..." made me expect the sentence to be contrasting aerobic and anaerobic conditions. Please reword the whole section to clearly convey the desired meaning.

3. RT-PCR measures transcript levels, not transcription (not a minor difference, as there is ample evidence for post-transcriptional regulation of transcript levels, including in circadian oscillations): 260-261: reduced gene transcription, we performed qRT-PCR analysis of a few representative genes of the nif cluster.

263- Analysis of transcription of genes involved in respiration,

4. Line 430 (and following sentences)- Metabolic state of the cell has been shown to be a major trigger of the internal clock in various systems

What is meant by trigger of the clock? Do the authors mean entraining cue? This is a very confusing terminology that is used two more times in the next couple of sentences with different implied meanings. The clock isn't "triggered" by anything, but is self-sustaining. However, its phase is entrained by metabolism. To say that the clock "triggers" a process would be unusual usage but maybe OK if explained better. The use in line 432 seems appropriate, but is not the same context as line 430. Again confusing in line 434: "However, even if triggered by metabolic cues, deletion of kaiA..."; deletion of KaiA is certainly not triggered by metabolic cues. The paragraph needs careful rewording to make sure the intended meaning is explained well in each sentence.

5. Lines 460-464: The expanded sentence is hard to follow. I suggest: ...enforcing segregation of the processes; however, in the...

6. Line 478: Synechococcus 7942 (needs the collection name or the number is not useful)

Lines 492-499 repeat too much of the Results without a punchline. The authors should more concisely set up the idea that analysis of the double mutant might help to sort out the source of residual rhythms in the kaiB1 mutant.

7. Line 993: Both HBP and HP used as abbreviations

8. General comment for the authors: the circadian community has developed and made available software packages for time series analysis from a variety of types of data that can be used by students and postdocs who aren't skilled at modeling. These can be identified from the methods and acknowledgment sections of circadian papers that use different model systems.

REVIEWER COMMENTS

Reviewer #1 (Remarks to the Author):

Review report for NCOMMS-23-08415A
January 11, 2024

This paper is a revised version of the paper submitted in March. Bandyopadhyay et al. showed the importance of the circadian clock in nitrogen-fixing growth in the unicellular diazotrophic cyanobacterium *Cyanothece* sp. ATCC 51142, which has not been able to be analyzed in a molecular biological aspect due to the lack of gene manipulation technique. As the authors emphasized in the response letter, this paper demonstrated that the circadian clock is important for nitrogen fixing growth under aerobic conditions in *Cyanothece* sp. ATCC 51142 using the *kaiA*-disrupted mutant for the first time. This reviewer appreciates this point. However, Reviewer 1 still felt that the paper tended to overstate the experimental results in many places, which is included in the comments below:

Major comments

1. The title has been revised by the addition of “component of the circadian clock”. However, this reviewer feels that this title itself is somewhat an exaggeration and would like to suggest the following title:

Oxygen dynamics produced by the circadian Kai clock is essential for maximal aerobic nitrogen-fixing growth in unicellular diazotrophic cyanobacterium *Cyanothece* sp. ATCC 51142

We appreciate the reviewer’s efforts in helping us improve this manuscript. We have taken into consideration the comments from both reviewers and have changed the title of the manuscript accordingly. The new title also meets the requirements of the journal (15 words max).

2. In Figure 3, the authors show that nitrogenase activity under anaerobic conditions is higher than under aerobic conditions, indicating that oxygen dynamics have a significant effect on nitrogenase activity. The acetylene reduction activity of nitrogenase is competitively inhibited by N₂. Therefore, the increase in activity on the argon sample compared to simply Air (i.e., 80% N₂) sample will include both an increase in activity with argon due to the loss of competitive inhibition by N₂ and an increase in activity with argon due to the relaxation of oxygen-induced inactivation of nitrogenase under anaerobic conditions. To clarify this, a comparison of activity with 20% oxygen-argon and 100% argon may be necessary.

Cellular oxygen dynamics does have a significant effect on nitrogenase activity. This has been extensively investigated and reported in cyanobacteria. Irrespective of the effect of N₂ on acetylene reduction assays, increasing concentrations of oxygen have been shown to reduce nitrogenase activity in unicellular nitrogen fixing cyanobacteria (Compaoré, J. and

Stal, L.J.: 2010, Oxygen and the light–dark cycle of nitrogenase activity in two unicellular cyanobacteria. Environmental Microbiology, 12: 54-62) shown that the nitrogenase cluster of Cyanothece 51142 is susceptible to increasing concentrations of oxygen (Liu et al., 2018, cited in the manuscript). In addition, nitrogenase mediated hydrogen production studies (that do not rely on acetylene reduction) in Cyanothece 51142 also revealed the oxygen sensitivity of its nitrogenase enzyme (Bandyopadhyay et al. 2010; cited in the manuscript). In the present study we show that under aerobic incubations, the integrity of the nitrogenase enzyme is lost. Molecular nitrogen present in air does not have such a destructive effect on the nitrogenase enzyme. Even though we agree with the reviewer that the increase in the nitrogenase activity under anaerobic incubation conditions is also influenced by lack of N₂ in the incubation space, such an experiment will not add any additional information or change the findings of this study. The manuscript highlights that i) when incubated in air, the Δ kaiA mutant grown under CL does not fix nitrogen efficiently, ii) when incubated in argon the inhibitory effect is eliminated and iii) the nitrogenase enzyme is degraded in samples incubated in air. These data lead us to conclude that under aerobic conditions, KaiA is necessary for N₂ fixation. These findings are independent of the effect of N₂ on the assays.

3. Given that nitrogenase activity is higher under anaerobic conditions than WT, this reviewer would like to see nitrogen-fixing growth under anaerobic conditions in addition to the comparison of nitrogenase activity (Figure 3).

Cyanobacteria, oxygenic photosynthetic microbes, cannot be grown under anaerobic conditions. In order to grow Cyanothece 51142 under anaerobic conditions, we would need to stop its photosynthetic activity completely. Although Cyanothece cells grow well under photomixotrophic and to a much lesser extent under photoheterotrophic conditions, it is unable to grow in complete darkness. Under photoheterotrophic conditions, photosystem II is active, evolving oxygen. Therefore, anaerobic growth of Cyanothece 51142 is not possible and cannot be documented.

4. The subtitle of line 251 is overstated. How about the following?: “Instability of nitrogenase in Δ kaiA”

We thank the reviewer for the suggestion. We have changed the subtitle to the one suggested. (line 284)

Minor points

5. Line 106, Please show the locus tags (cce_XXXXX) for kaiA, kaiB1, and kaiC genes. *We have provided the locus tags for the three genes.(lines 113-114)*

6. Typo, kaiB1 and kaiC1 are written as kaiB and kaiC, respectively. (lines 109-117, 349, 355...)

We have changed kaiB and kaiC to B1 and C1 wherever relevant.

7. Line 123-125, please clearly describe that growth is nitrogen-fixing growth (photosynthesis + nitrogen fixation). This important information is just shown as “in nitrogen deficient media” in the legend in Figure 1.

We have clearly stated in the results section that growth was assessed under nitrogen fixing and photoautotrophic growth conditions. (lines 134-135)

8. Description of the result of Figure 1 (lines 126-127), The authors simply describe “the WT exhibits an endogenous rhythm”, but what is observed in the growth curve is not an endogenous rhythm, but rather a staircase-like growth. This reviewer feels the following expression should be better: Cells grow during subjective light period (increase in OD730) and do not grow during subjective dark period (constant or decrease in OD730), and these repetitions create an apparent endogenous rhythm in growth.

Endogenous rhythm is used to describe biological processes which alter periodically, although external conditions remain constant. In our study we show that the rhythms originate and sustain under constant light conditions. There is no subjective dark period per say as the cultures have never been exposed to LD cycles. The rhythm that is enforced by growing cells under LD cycles and then transitioning them to CL is different from this natural rhythm that the cells exhibit under CL growth. However, we do agree that this can be confusing to readers and we have explained the rhythm in our results section by stating the concept of rise and fall in OD 730 corresponding to the photosynthetic and nitrogen fixing phases of the cells (lines 137-139). Originally, we had this information in the discussion section alone.

9. Fig. S8, panels C and D, Phycocyanin and chlorophyll contents are shown as Fold change (WT/ Δ kaiA), but should be Δ kaiA/WT if WT is the standard. In addition, no explanation is given for the color indication on the right sides of panels C and D.

We have followed the reviewer’s suggestion and presented the phycocyanin and chlorophyll content as Δ kaiA/WT. We have labeled the color panels in the figure and explained them.

10. Figure 3, Nitrogenase activity is shown as relative values, but the actual values should be indicated somewhere.

The actual nitrogenase activity measured in the WT and the kaiA mutant under the assay conditions reported in this study have been included as a supplementary table (Table S1).

11. Figure 4, Vertical axis says OD720, but it should be OD730, as described in the text.

We have changed OD720 to OD730 to keep it consistent.

12. Figure S17, The description in the legend “nitrogen deficient media” does not match the description in the text, lines 391-392 “under nitrogen sufficient conditions”. Also, the information on light conditions (when LD change to CL) is not stated.

Thank you for pointing this out. We have made the correction. We have also added bars and arrows to indicate the LD conditions and the shift to CL.

14. Figure 5, In the Western blot analysis, only NifD was detected even though they used an anti-NifDK antibody. Please explain this point briefly. Also, the description regarding expression and purification of NifD protein that was used as a marker should be described in the Materials and Methods section.

The anti-Nif antibody used in this work was originally developed by Prof. Paul Ludden (Grunwald et al., 1995) . It was later used and referred to as an anti-NifDK antibody by Prof. Louis Sherman in studies in Cyanothecae (Colon-Lopez et al., 1997). Subsequent western blot analysis comparing purified NifD protein and total cellular protein from nitrogen fixing Cyanothecae cells (Liu et al., 2018) showed that this antibody is specific to NifD. We realize that this can be confusing and therefore instead of using the original name, we have referred to it as the anti-NifD antibody. The expression and purification of the NifD protein has been reported earlier and we have included the citation in the methods section of our manuscript (lines 608,609).

We sincerely thank the reviewer for carefully reviewing the manuscript and for suggestions and comments to help improve it.

Reviewer #2 (Remarks to the Author):

The revised manuscript is greatly improved, and addresses most of the concerns of the reviewers. A few problems with presentation remain.

1. The title still suggests that there is a special role for KaiA in oxygen dynamics that is different from a role in completing the circadian clock mechanism. An accurate title would be: KaiA deletion in a unicellular diazotrophic cyanobacterium implicates the circadian clock in regulating intracellular oxygen dynamics

We thank the reviewer for the suggestion. Taking into consideration the comments from both reviewers, we have modified the title of the manuscript. The new title also meets the requirements of the journal (15 words max).

2. Lines 203-206:

When cells grown under 12:12h LD cycles were incubated under aerobic conditions, the average specific rates of nitrogenase activity in the Δ kaiA mutant was almost similar to that

of the WT, whereas mutant cells grown under 16:8h LD cycles exhibited nitrogen fixation rates of up to 80 to 90 % that of the WT (Fig. 3a,b).

A few things about this wording are confusing. “almost similar to” doesn’t make sense. They were almost identical, or they were similar? And this result is contrasted (whereas) with a condition that was 90% the same as WT. How is that not “almost similar”? Think about what point you want to make and be sure to say it clearly – do you want to show how findings are alike, or point out differences? Also, the lead in with “When cells grown...were incubated under aerobic conditions...” followed by “whereas...” made me expect the sentence to be contrasting aerobic and anaerobic conditions. Please reword the whole section to clearly convey the desired meaning.

We do agree that the section was confusing and we have incorporated the changes suggested by the reviewer.(lines 194-203)

3. RT-PCR measures transcript levels, not transcription (not a minor difference, as there is ample evidence for post-transcriptional regulation of transcript levels, including in circadian oscillations):

260-261: reduced gene transcription, we performed qRT-PCR analysis of a few representative genes of the nif cluster.

263- Analysis of transcription of genes involved in respiration,

We agree with the reviewer. We have made the necessary changes.(lines 205, 208, 336)

4. Line 430 (and following sentences)- Metabolic state of the cell has been shown to be a major trigger of the internal clock in various systems

What is meant by trigger of the clock? Do the authors mean entraining cue? This is a very confusing terminology that is used two more times in the next couple of sentences with different implied meanings. The clock isn’t “triggered” by anything, but is self-sustaining. However, its phase is entrained by metabolism. To say that the clock “triggers” a process would be unusual usage but maybe OK if explained better. The use in line 432 seems appropriate, but is not the same context as line 430. Again confusing in line 434: “However, even if triggered by metabolic cues, deletion of kaiA...”; deletion of KaiA is certainly not triggered by metabolic cues. The paragraph needs careful rewording to make sure the intended meaning is explained well in each sentence.

We agree with the reviewer. We have revised the discussion section to avoid confusion with the usage of the word “trigger”. (lines 345-50)

5. Lines 460-464: The expanded sentence is hard to follow. I suggest:
...enforcing segregation of the processes; however, in the...

We thank the reviewer for the suggestion. We have made the change in the manuscript.(line 373)

6. Line 478: *Synechococcus* 7942 (needs the collection name or the number is not useful)

We have added the collection name for Synechococcus 7942 (line 391)

Lines 492-499 repeat too much of the Results without a punchline. The authors should more concisely set up the idea that analysis of the double mutant might help to sort out the source of residual rhythms in the *kaiB1* mutant.

We have eliminated the repetitions and have indicated the importance of future analysis of the double mutant in understanding the residual rhythms in $\Delta kaiA$. (lines 394-98).

7. Line 993: Both HBP and HP used as abbreviations

We have made the corrections. Thanks for pointing it out. (lines 830, 31)

8. General comment for the authors: the circadian community has developed and made available software packages for time series analysis from a variety of types of data that can be used by students and postdocs who aren't skilled at modeling. These can be identified from the methods and acknowledgment sections of circadian papers that use different model systems.

That is great to know. Thank you for bringing this to our attention. We will explore and include these in our future analyses of the clock mutants.

Once again, we are grateful to the reviewer for all the suggestions and comments that greatly helped improve the manuscript.

Reviewer #1 (Remarks to the Author):

Review report for NCOMMS-23-08415B
February 25, 2024

This paper was originally submitted last March, resubmitted in December, and is a re-revised version with further revisions. During the two revision processes, the authors, Bandyopadhyay et al., made revisions with input from two reviewers. Reviewer 1 has no further revision comments and believes that the paper is now sufficient to be published in Nature Communication.

Reviewer #2 (Remarks to the Author):

The authors have addressed the remaining issues from the last review.